# Is Routine Screening Using Duplex Ultrasonography for Deep Vein Thrombosis Necessary after Shoulder Arthroplasty?

**DOI:** 10.3390/diagnostics13040636

**Published:** 2023-02-08

**Authors:** Du-Han Kim, Sang-Soo Na, Ui-Jun Park, Chul-Hyun Cho

**Affiliations:** 1Department of Orthopaedic Surgery, Keimyung University Dongsan Hospital, Keimyung University School of Medicine, 1035 Dalgubul-ro, Dalseo-gu, Daegu 42601, Republic of Korea; 2Department of Vascular Surgery, Keimyung University Dongsan Hospital, Keimyung University School of Medicine, 1035 Dalgubul-ro, Dalseo-gu, Daegu 42601, Republic of Korea

**Keywords:** shoulder, arthroplasty, deep vein thrombosis, venous thromboembolism, duplex ultrasonography

## Abstract

The aims of this study were to examine the incidence, risk factors, and outcomes of deep vein thrombosis (DVT) after shoulder arthroplasty in Korean patients. A total of 265 patients who underwent shoulder arthroplasty were included. The mean age of the patients was 74.6 years, and there were 195 females and 70 males. Clinical data including patient demographics, blood tests, and previous and current medical history were investigated. For screening of DVT, duplex ultrasonography of the operative arm was performed 2 to 5 days after surgery. Of the 265 patients, 10 patients (3.8%) were diagnosed with DVT using postoperative duplex ultrasonography. There were no cases of pulmonary embolism. There were no significant differences between the DVT and no DVT groups regarding all clinical data, except for the Charlson comorbidity index (CCI), which was significantly higher in the DVT group than in the no DVT group (5.0 vs. 4.1; *p* = 0.029). All patients had asymptomatic DVT that showed complete resolution after administration of antithrombotic agents or close observation without medications. The overall incidence of DVT was 3.8% during a period of 3 months after shoulder arthroplasty in Korean patients, and most cases were asymptomatic. Routine screening for DVT using duplex ultrasonography after shoulder arthroplasty may not be necessary except in patients with high CCI.

## 1. Introduction

Deep vein thrombosis (DVT) is thought to be causally related to three factors that comprise the Virchow triad: hypercoagulability, venous stasis, and endothelial injury [1]. DVT after surgery is a critical complication that can be life-threatening when it leads to pulmonary embolism (PE) [1,2,3]. Therefore, prevention of DVT may be a critical safety concern for patients undergoing orthopedic surgery [4,5].

DVT after total hip arthroplasty or total knee arthroplasty is common and well established for its incidence, risk factors, treatment, and prevention [6,7,8]. However, DVT after shoulder arthroplasty has been regarded as a relatively rare complication, and there is still controversial regarding its precise incidence [2,8]. Recently, there have been increasing reports of DVT or PE associated with shoulder arthroplasty [2]. Previous studies have reported that the incidence of venous thromboembolism (VTE) including DVT or PE after shoulder arthroplasty ranges from 0% to 14% [9,10,11,12,13,14,15,16]. According to a recent systematic review, the incidence of DVT after shoulder arthroplasty was 0.54%, and the incidence of pulmonary embolism was 0.33%, with an overall incidence of VTE of 0.81% [17].

Although diagnosis of DVT is challenging as no physical examination findings are sensitive or specific, clinical suspicion should be high for patients with unilateral limb swelling, redness, pain, or a palpable cord [3]. Several studies have suggested that the incidence of DVT after shoulder surgery may be underestimated because the patients with DVT are generally asymptomatic [4,9]. According to a review of the literature, the low incidence rate of DVT might be attributed to the fact that most previously reported studies included retrospective case series that described symptomatic DVT [9,11,12,13,14,16]. However, a recent prospective study by Willis et al. [15] reported that the incidence of DVT confirmed by Doppler ultrasonography after shoulder arthroplasty was 13%, comparable to that after total hip arthroplasty.

DVT can lead to a significant increase in morbidity and mortality, placing a substantial burden on public health [17]. Considering asymptomatic or undiagnosed DVT, along with an exponential increase in cases of anatomical total shoulder arthroplasty (ATSA) or reverse total shoulder arthroplasty (RTSA), increased incidence of DVT after index surgery may be inevitable and, therefore, problematic [8,9,18,19]. Due to the potential for development of fatal PE, early detection of DVT is important [3]. Duplex ultrasonography is regarded as a useful tool for screening and diagnosis of DVT for the upper or lower extremities [9]. Numerous studies have demonstrated the usefulness of duplex ultrasonography in the detection and location of postoperative DVT after extremity surgery [9,13,14,15].

Whereas previous studies on the incidence and risk factors of DVT after shoulder arthroplasty were performed in Western populations, studies conducted in Asian populations have rarely been reported. The incidence of silent or asymptomatic DVT in the early postoperative period after shoulder arthroplasty in Asian populations is unknown. The aims of this study were to examine the incidence, risk factors, and outcomes of DVT after shoulder arthroplasty in Korean patients and to evaluate the role of postoperative duplex ultrasonography.

## 2. Materials and Methods

This retrospective study was conducted on the basis of prospectively collected data after obtaining approval from the Keimyung University Dongsan Hospital Institutional Review Board (IRB No: 202212029). Between January 2014 and March 2021, 265 patients who underwent shoulder arthroplasty and postoperative duplex ultrasonography (Logiq E9, GE Healthcare, Wauwatosa, WI, USA) for detection of DVT were investigated for this study. In patients who underwent shoulder arthroplasty, RTSA was performed in 229 cases, hemiarthroplasty was performed in 21 cases, and ATSA was performed in 15 cases. Regarding indications for shoulder arthroplasty, 210 patients had various shoulder diseases (79 cuff tear arthroplasty, 58 irreparable massive rotator cuff tear, 45 degenerative glenohumeral osteoarthritis, 12 infection control surgery state due to shoulder infection, five recurrent shoulder dislocation, four osteonecrosis of the humeral head, four RTSA state with complication, one ATSA state with complication, one hemiarthroplasty state with complication, and one rapid destructive glenohumeral arthrosis) and 55 patients had proximal humerus fractures (48 acute proximal humerus fracture and nine proximal humerus old fracture with sequalae). The mean age of the patients was 74.6 years (range 48–98 years), and there were 195 females and 70 males. The right shoulder was involved in 184 patients, and the left shoulder was involved in 81 patients.

Clinical data, including body mass index (BMI), smoking, previous history of ipsilateral shoulder surgery, operative time, and hospitalization period, were recorded. Blood tests including complete blood count (white blood cells and platelets), hemoglobin, coagulation profile (prothrombin time and activated prothrombin time), and renal function test (serum creatinine and estimated glomerular filtration rate) were also recorded. Previous and current medical history including use of antithrombotic agents, history of VTE, asthma, cardiac arrhythmia, diabetes, liver cirrhosis, cerebrovascular accident (CVA), ischemic heart disease (IHD), or malignancy was recorded. The Charlson comorbidity index (CCI) was calculated for prediction of mortality in patients with comorbidities. Sixty-five patients used antithrombotic agents (aspirin, clopidogrel, warfarin, etc.) because of medical comorbidities. Antithrombotic agents were discontinued at 3 or 5 days prior to index surgery.

All operative procedures were performed by a single surgeon. The operation was performed with the patient in the beach chair position under general anesthesia using a deltopectoral approach. None of the patients received perioperative DVT prophylaxis including an anti-embolic stocking, pneumatic compression device, or administration of intraoperative heparin. Wearing an abduction brace, all patients engaged in range-of-motion exercises of the elbow, wrist, and hand immediately after surgery. Beginning 1 week after surgery, passive range of motion exercise of the shoulder was initiated and individualized according to the disease entity. Monitoring for symptoms (pain, cramping, soreness, feeling of warmth of limb, sudden shortness of breath, or chest pain) and signs (swelling, color change of limb, rapid or irregular heartbeat, dizziness, excessive sweating, or fever) of VTE including DVT or PE was performed for 3 months after index surgery.

Duplex ultrasonography of the operative arm was performed between 2 and 5 days after index surgery for evaluation of the incidence of DVT in all patients. The examination was performed by an experienced sonographer certificated by the American Registry for Diagnostic Medical Sonography and interpretation was performed by an experienced vascular surgeon. The patient underwent duplex ultrasonographic examination in supine and arm abduction position using a 4–14 MHz linear probe for the cephalic, basilic and brachial veins and a 3–11 MHz linear probe for the axillar and subclavian veins. The presence, location, and nature of thrombosis was assessed and recorded. Echogenic material in the lumen of the deep vein, incompressible vein walls with external transducer pressure, no detectable vein wall motion in central veins in a B-mode image, and color filling defects on color Doppler ultrasonography were regarded as positive findings of acute thrombosis.

### Statistical Analysis

Statistical analysis was performed using the SPSS statistical package (version 25.0; IBM, Armonk, NY, USA). Linear logistic regression analysis was performed to identify potential risk factors of DVT. Significant differences in the variables of the no DVT and DVT groups were identified using the Mann–Whitney U test for continuous variables and the chi-squared test for categorical data. Statistical significance was accepted for *p*-values of <0.05.

## 3. Results

Of 265 patients, 10 patients (3.8%) were diagnosed with DVT using postoperative duplex ultrasonography. There were no cases of PE. According to the results of linear logistic regression analysis, development of DVT showed a significant association with CCI (4.1 vs. 5.0; *p* = 0.029). No significant differences in regard to age, sex, side, operative indication, type of arthroplasty, BMI, white blood cell count, hemoglobin, platelet, international normalized ratio, active partial thromboplastin time, serum creatinine, estimated glomerular filtration rate, smoking, use of antithrombotic agent, VTE history, and previous and current medical history (asthma, cardiac arrhythmia, diabetes mellitus, liver cirrhosis, cerebrovascular accident, ischemic heart disease, malignancy, or previous shoulder surgery), and operative time were observed between the no DVT and DVT groups (*p* > 0.05) (Table 1).

All patients had asymptomatic DVT. According to the findings of duplex ultrasonography, eight patients were prescribed antithrombotic treatments including rivaroxaban, enoxaparin, warfarin, or combinations by a vascular surgeon. Two patients had closed observation without antithrombotic treatment. All patients had complete resolution without any complications.

### Case Presentation

A 76 year old male presented with painful limitation of motion of the right shoulder. He had a displaced three-part proximal humerus fracture resulting from a slip down. Medical comorbidities included chronic kidney disease, cardiac arrhythmia, and diabetes. The CCI was 7. The patient had no history of hematologic clotting or bleeding abnormalities. The RTSA procedure was performed at 11 days after the initial trauma with an operative time of 115 min. Two days after RTSA, acute brachial vein thrombosis was detected by duplex ultrasonography of the right upper extremity. However, there was no symptoms for DVT. Enoxaparin 20 mg (Xarelto; Bayer Schering Pharma, Berlin, Germany) and warfarin sodium 2 mg (Warfarin; Daehwa Phamaceutical, Seoul, Republic of Korea) were administered for 2 weeks. Warfarin sodium 2 mg was then administered over the next 4 weeks was administered. No complications related to DVT were detected during the serial follow-up evaluation (Figure 1).

## 4. Discussion

According to the findings of the current study, the incidence of DVT was 3.8% in the early postoperative period after shoulder arthroplasty in Korean patients. High CCI showed an association with the development of DVT. All patients had asymptomatic DVT that showed complete resolution after administration of antithrombotic agents or close observation without medications. These findings from the current study revealed that routine screening for DVT using duplex ultrasonography after shoulder arthroplasty may not be necessary except in patients with high CCI.

Numerous studies have demonstrated that shoulder arthroplasty is a useful treatment strategy for various shoulder conditions including rotator cuff arthropathy, glenohumeral arthritis, and proximal humerus fractures [20,21]. According to the study reported by Best et al. [18], there has been a dramatic increase in trends regarding the incidence of shoulder arthroplasty. In particular, the incidence of annual procedures of primary RTSA nearly tripled from 2012 to 2017 [18]. From 2012 to 2017, the population-adjusted incidence of primary RTSA increased from 7.3 cases per 100,000 persons (22,835 procedures) to 19.3 cases per 100,000 (62,705 procedures), that of anatomic TSA increased from 9.5 cases per 100,000 (29,685 procedures) to 12.5 cases per 100,000 (40,665 procedures), and that of hemiarthroplasty decreased from 3.7 cases per 100,000 (11,695 procedures) to 1.5 cases per 100,000 (4930 procedures) [18]. Although reports of postoperative DVT have also increased along with the exponential growth of shoulder arthroplasty, a consensus regarding DVT after shoulder arthroplasty including its incidence, risk factors, treatment, and prophylaxis is still not well established.

Previous studies have reported that the incidence of VTE including DVT or PE after shoulder arthroplasty ranges from 0% to 14% [9,10,11,12,13,14,15,16]. Kolz et al. [13] reported that the overall incidence of symptomatic VTE after shoulder arthroplasty was 0.41% (15 VTEs/5906 cases) in a single institution over a 16 year period. They concluded that the risk of VTE after shoulder arthroplasty is low, and that routine use of perioperative chemoprophylaxis may not be necessary [13]. By contrast, a prospective study conducted by Willis et al. [15] reported a high incidence of VTE of 14% (14/100) after shoulder arthroplasty. They suggested that shoulder surgeons should pay attention to the potential risk of VTE until 3 months after index surgery because this value is comparable to that after total hip arthroplasty [15]. Kunutsor et al. [7] investigated 43 studies with ICD code data on 672,495 ATSAs. They reported that the overall incidence of VTE during a 3 month period after TSA was 0.85%. In a systematic review reported by Na et al. [17], the overall incidence of VTE after shoulder arthroplasty was 0.81% (78/9681), showing a wide range from 0% to 14%. The incidence of DVT was 0.54% (52/9681), and that of PE was 0.33% (42/12566). There were no significant differences in the incidences according to the type of arthroplasty. The lowest value denotes VTE events with significant symptoms that required readmission for management, typically reported in the retrospective studies [8]. The highest value denotes VTE events including asymptomatic DVTs confirmed by duplex ultrasonographic screening of all subjects in a prospective cohort study [8]. In the current study, the incidence of DVT diagnosed by duplex ultrasonography in the early postoperative period after shoulder arthroplasty was 3.8%. There were no cases of PE. Although the results of our study cannot be generalized due to the small sample size, the incidence of DVT in Korean patients who underwent shoulder arthroplasty might be somewhat low compared to that reported in Western patients. In particular, all patients were asymptomatic. Takahashi et al. [22] reported an overall incidence of DVT after elective arthroscopic shoulder surgery was 5.7% (10/175). They found that all patients were asymptomatic; however, development of an asymptomatic pulmonary embolus occurred in one patient within 3 months after index surgery [22]. Although the clinical relevance regarding the incidence of asymptomatic DVT is still controversial, there is no doubt that symptomatic DVT and fatal PE may originate from asymptomatic DVT.

Kunutsor et al. [7] reported that VTE after shoulder arthroplasty reaches a peak at 60 days in the postoperative period and then shows a gradual decline thereafter. Ojike et al. [2] reported that VTE after shoulder surgery may be underreported and recommended the postoperative duplex ultrasonography screening of upper and lower extremities in high-risk patients. Suspicion of silent VTE may be required for up to 3 months after index surgery. There is no information regarding the optimal timing of duplex ultrasonographic screening for postoperative DVT. However, the hypercoagulable state after surgical injury persists for as long as 48 h, during which a peak for the development of DVT is reached [9]. In the current study, duplex ultrasonography of the operative arm was performed between 2 and 5 days after index surgery in all patients. Monitoring for symptoms and signs of VTE including DVT or PE was performed during 3 months after index surgery.

The general risk factors of DVT for orthopedic surgeries including upper and lower limbs are similar [9] and include a history of VTE, old age, obesity, smoking, diabetes mellitus, history of malignancy, longer operation time, and venous stasis [9]. Specific risk factors of developing DVT in upper-limb surgeries included the presence of a central venous catheter, operating in the lateral decubitus position with the affected limb in traction, and interscalene block [9]. In a systematic review reported by Na et al. [17], the risk factors for VTE included age over 70 years, higher BMI, higher CCI, history of VTE, asthma, cardiac arrhythmia, diabetes, lower hemoglobin level, use of general endotracheal anesthesia without interscalene nerve block, traumatic indication, longer operative time, and revision arthroplasty. Dattani et al. [5] reported that diabetes mellitus, rheumatoid arthritis, and ischemic heart disease were identified as the major risk factors of VTE after shoulder surgery. In the current study, there were no significant differences between DVT and no DVT groups regarding all clinical data, except that the CCI in DVT group was significantly higher than that in the no DVT group. Furthermore, all patients had asymptomatic DVT that showed complete resolution after administration of antithrombotic agents or close observation without medications. These findings suggest that routine screening using duplex ultrasonography for DVT may not be necessary after shoulder arthroplasty except in patients with high CCI. In addition, quantification of the risk factors for DVT after shoulder arthroplasty is necessary to determine whether DVT prophylaxis is needed.

The current study had several limitations. First, duplex ultrasonography of the operative arm was only performed between postoperative days 2 and 5 in all patients. Duplex ultrasonographic examinations of the contralateral arm and both lower extremities were not performed. Moreover, no examinations were performed during the subacute phase (3 weeks to 3 months after index surgery). However, monitoring for symptoms and signs of VTE including DVT or PE was performed for 3 months after index surgery. Second, the sample size was too small to identify potential risk factors of DVT after shoulder arthroplasty. However, this is the first cohort study to assess the incidence and risk factors of DVT after shoulder arthroplasty in an Asian population, which is meaningful. Further prospective multicenter studies with a well-designed protocol are warranted to clarify the precise incidence, potential risk factors, treatment, and prophylaxis of DVT after shoulder arthroplasty.

## 5. Conclusions

The overall incidence of DVT was 3.8% during a period of 3 months after shoulder arthroplasty in Korean patients, and most cases were asymptomatic. Routine screening for DVT using duplex ultrasonography after shoulder arthroplasty may not be necessary except in patients with high CCI.

## Figures and Tables

**Figure 1 diagnostics-13-00636-f001:**
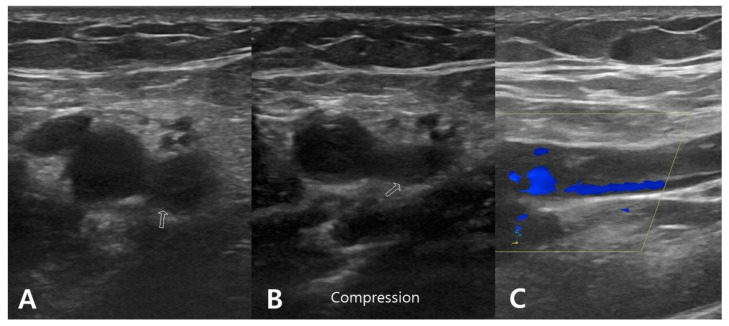
Duplex ultrasonographic image from a 76 year old man who underwent reverse total shoulder arthroplasty for proximal humerus fracture. (**A**,**B**) B-mode transverse image showing hypoechoic thrombus (arrow) in the brachial vein, which is not compressible by the probe. (**C**) Longitudinal color Doppler image revealing color filling defect in the brachial vein.

**Table 1 diagnostics-13-00636-t001:** Estimated association for potential risk factors of deep vein thrombosis after shoulder arthroplasty.

	No DVT (*n* = 255)	DVT (*n* = 10)	*p*-Value
Age (years)	74.5 ± 6.8	76.3 ± 4.2	0.462
Sex (male/female)	67/188	3/7	0.793
Side (right/left)	177/78	7/3	1.000
Indication (disease/trauma)	204/51	6/4	0.126
Type of arthroplasty (RTSA/TSA/HA)	221/14/20	8/1/1	0.798
BMI	24.6 ± 3.9	24.7 ± 4.6	0.833
WBC count	7.1 ± 2.8	8.1 ± 3.4	0.332
Hb	12.2 ± 1.8	12.1 ± 2.4	0.807
Platelet	244.2 ± 82.1	279.4 ± 86.7	0.161
INR	1.0 ± 0.2	1.3 ± 1.0	0.072
aPTT	26.6 ± 6.6	29.9 ± 10.4	0.347
sCr	0.9 ± 0.7	2.5 ± 3.1	0.061
e-GFR	81.6 ± 27.9	58.8 ± 37.8	0.066
Smoking (no/yes)	229/26	10/0	0.605
Use of antithrombotic agent (no/yes)	191/64	9/1	0.459
VTE history (no/yes)	250/5	10/0	1.000
Asthma (no/yes)	239/16	10/0	1.000
Cardiac arrhythmia (no/yes)	216/39	7/3	0.212
Diabetes mellitus (no/yes)	181/74	8/2	0.729
Previous shoulder surgery (no/yes)	211/44	10/0	0.377
Liver cirrhosis (no/yes)	250/5	10/0	1.000
CVA (no/yes)	220/35	7/3	0.150
IHD (no/yes)	225/30	9/1	1.000
Malignancy (no/yes)	231/24	9/1	0.950
CCI	4.1 ± 1.5	5.0 ± 1.3	0.029 *
Operative time	94.0 ± 32.1	89.2 ± 18.3	0.913

DVT, deep vein thrombosis; RTSA, reverse total shoulder arthroplasty; TSA, total shoulder arthroplasty; HA, hemiarthroplasty; BMI, body mass index; WBC, white blood cell; INR, international normalized ratio; aPTT, active partial thromboplastin time; sCr, serum creatinine; e-GFR, estimated glomerular filtration rate; VTE, venous thromboembolism; CVA, cerebrovascular accident; IHD, ischemic heart disease; CCI, Charlson comorbidity index; * statistically significant.

## Data Availability

Not applicable.

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
