# Peer review of "Is Routine Screening Using Duplex Ultrasonography for Deep Vein Thrombosis Necessary after Shoulder Arthroplasty?"

_diagnostics, 2023, doi:10.3390/diagnostics13040636_

Round 1
Reviewer 1 Report
The manuscript is well-written and it should be published.
Author Response
The manuscript is well-written and it should be published.
-> Thank you very much.
Reviewer 2 Report
The reference list of the manuscript contains 22 titles, and is without inappropriate self-citations. None reference is elder than 20 years. The manuscript is clear, with a high rate of clinical significance. The manuscript present scientific resound and the design appropriate to test the hypothesis. The methods have to be more cleary described. All aspects regarding the figures/images are appropriate, and they are easy to interpret and understand. The presentation and the analyzed date are written in proper way. The presentation of the results are at high standard, with appropriate statistics.
But I've still some comments:
1. Please explain abbreviation PE in line 18.
2. Please add further information regarding the monitoring during the follow up period up to 3 months. Lines 90-93.
3. You should add more detailed information about the onset of DVT after shoulder arthroplasty. And please explain to the reader why you performed Doppler ultrasound only during the first 5 days. LL 112-115 & 203-205.
4. LL 136: arr"h"ythmie --> missing h
5. Can you give sone date to the audience regarding the incidence of PE in the Non DVT group?
Author Response
The reference list of the manuscript contains 22 titles, and is without inappropriate self-citations. None reference is elder than 20 years. The manuscript is clear, with a high rate of clinical significance. The manuscript present scientific resound and the design appropriate to test the hypothesis. The methods have to be more cleary described. All aspects regarding the figures/images are appropriate, and they are easy to interpret and understand. The presentation and the analyzed date are written in proper way. The presentation of the results are at high standard, with appropriate statistics.
But I've still some comments:
- Please explain abbreviation PE in line 18.
--> Thanks for your comment. According to your comment, we changed to “pulmonary embolism”
- Please add further information regarding the monitoring during the follow up period up to 3 months. Lines 90-93.
--> Thanks for your comment. According to your comment, we changed to “Monitoring for symptoms (pain, cramping, soreness, or feeling of warmth of limb, sudden shortness of breath, chest pain) and signs (swelling, color change of limb, rapid or irregular heartbeat, dizziness, excessive sweating, fever) of VTE including DVT or PE was performed during 3 months after index surgery.”
- You should add more detailed information about the onset of DVT after shoulder arthroplasty. And please explain to the reader why you performed Doppler ultrasound only during the first 5 days. LL 112-115 & 203-205.
--> Thanks for your comment. We clarified them in the Discussion section. “Kunutsor et al. [7] reported that VTE after shoulder arthroplasty reaches a peak at 60 days postoperative period and then shows a gradual decline thereafter. Ojike et al. [2] reported that VTE after shoulder surgery may be underreported and recommended that postoperative duplex ultrasonography screening of upper and lower extremities in high-risk patients. Suspicion of silent VTE may be required for up to 3 months after index surgery. There is no information regarding optimal timing of duplex ultrasonographic screening for postoperative DVT. However, the hypercoagulable state after surgical injury persists for as long as 48 hours, during which time will reach a peak for the development of DVT [9]. In the current study, duplex ultrasonography of the operative arm was performed between 2 and 5 days after index surgery in all patients. Monitoring for symptoms and signs of VTE including DVT or PE was performed during 3 months after index surgery.” For why performed Doppler ultrasound only during the first 5 days, we described them in the limitations of this study. “First, duplex ultrasonography of the operative arm was only performed between post-operative 2 and 5 days in all patients. Duplex ultrasonographic examinations of the contralateral arm and both lower extremities were not performed. And no examina-tions were performed during the subacute phase (3 weeks to 3 months after index sur-gery). However, monitoring for symptoms and signs of VTE including DVT or PE was performed during 3 months after index surgery.”
- LL 136: arr"h"ythmie --> missing h
--> Thanks for your comment. According to your comment, we changed to “arrhythmia”
- Can you give sone date to the audience regarding the incidence of PE in the Non DVT group? --> Thanks for your comment. As we mentioned in main-text, there were no cases of pulmonary embolism in our study.